# COVID-19 Vaccination in the WHO African Region: Progress Made in 2022 and Factors Associated

**DOI:** 10.3390/vaccines11051010

**Published:** 2023-05-22

**Authors:** Franck Mboussou, Bridget Farham, Sheillah Nsasiirwe, Ajiri Atagbaza, Daniel Oyaole, Phionah Lynn Atuhebwe, Victor Alegana, Fred Osei-sarpong, Ado Bwaka, Gilson Paluku, Amos Petu, Oniovo Efe-Aluta, Akpaka Kalu, Magaran Monzon Bagayoko, Benido Impouma

**Affiliations:** World Health Organization, Regional Office for Africa, Brazzaville P.O. Box 06, Congo

**Keywords:** African region, COVID-19 vaccines, coverage

## Abstract

This study summarizes progress made in rolling out COVID-19 vaccinations in the African region in 2022, and analyzes factors associated with vaccination coverage. Data on vaccine uptake reported to the World Health Organization (WHO) Regional Office for Africa by Member States between January 2021 and December 2022, as well as publicly available health and socio-economic data, were used. A negative binomial regression was performed to analyze factors associated with vaccination coverage in 2022. As of the end of 2022, 308.1 million people had completed the primary vaccination series, representing 26.4% of the region’s population, compared to 6.3% at the end of 2021. The percentage of health workers with complete primary series was 40.9%. Having carried out at least one high volume mass vaccination campaign in 2022 was associated with high vaccination coverage (β = 0.91, *p* < 0.0001), while higher WHO funding spent per person vaccinated in 2022 was correlated with lower vaccination coverage (β = −0.26, *p* < 0.03). All countries should expand efforts to integrate COVID-19 vaccinations into routine immunization and primary health care, and increase investment in vaccine demand generation during the transition period that follows the acute phase of the pandemic.

## 1. Introduction

Since the identification of the severe acute respiratory syndrome coronavirus 2 (SARS-CoV-2) in Wuhan, China, in 2020 [1], efforts to control the coronavirus disease 2019 (COVID-19), have had two main aims: reducing its circulation by protecting vulnerable individuals, especially those at risk of severe disease or occupational exposure to the virus, and; diagnosing and treating COVID-19-related disease to reduce mortality [2]. The accelerated development and approval of COVID-19 vaccines for emergency use led to vaccination being added to the response interventions less than one year after the pandemic began [3,4]. The first COVID-19 vaccine was delivered outside of a clinical trial setting on 8 December 2020, in the United Kingdom [5]. The vaccines were developed to generate an immune response to prevent the SARS-CoV-2 infection, or reduce the risk of severe disease or death and long-term sequelae [6]. The World Health Organization (WHO) Strategic Advisory Group of Experts on Immunization (SAGE) recommended four objectives for vaccination programs to achieve the overall goal of full recovery from the COVID-19 pandemic: (i) minimize deaths, severe disease, and overall disease burden; (ii) curtail the virus’ impact on health systems; (iii) resume socioeconomic activities in full, and; (iv) reduce the risk of new variants [7]. As vaccine supply was constrained, especially in 2021, the WHO SAGE recommended that countries prioritize vaccination of high-risk populations, including health workers, older adults, and people with comorbid conditions [7].

The African region is made up of 47 out of the 54 countries of Africa [8]. In the region, Seychelles was the first country to deploy COVID-19 vaccines, starting on 13 January 2021 [9]. Forty-six countries in the African region are rolling out COVID-19 vaccinations; the exception being Eritrea [10]. In October 2021, the WHO published an ambitious strategy to ensure that all countries had vaccinated 40% of their populations by the end of 2021, and 70% by mid-2022 [11]. At the end of December 2021, five countries had achieved the 2021 target (Seychelles, Mauritius, Rwanda, Botswana, and Cape Verde) and 26 countries were yet to reach 10% of their population with the complete primary vaccination series. In response, the WHO Regional Office for Africa (WHO AFRO) conducted a by-country risk assessment of slow vaccination uptake; the assessment prioritized 20 countries, most of which had not reached coverage of 10% of their population having completed the primary vaccination series by the end of December 2021 [10]. WHO AFRO deployed experts to these countries to support governments’ efforts to scale up COVID-19 vaccinations, as part of its Multi-Partners’ Country Support Teams (MP–CST) Initiative [10]. This initiative has helped targeted countries to diversify service delivery strategies, combining health facility fixed sites with mobile and outreach teams; to carry out mass vaccination campaigns; and to engage more community leaders in demand creation [10].

Factors influencing vaccination coverage in the African region have not been sufficiently documented. In high-income countries, apart from the availability of vaccines, early success of COVID-19 vaccination programs was driven by political commitment, the capacity of health systems to ensure large deployment of vaccines, the population’s trust in health authorities and governments, high COVID-19 risk perception, and confidence in the vaccine’s safety and effectiveness [12,13,14]. Tracking drivers of vaccine uptake and vaccination coverage in the African region could inform how WHO AFRO prioritizes actions aiming to ramp up COVID-19 vaccinations in the region given current low COVID-19 incidences and related death rates, which may lead to reduced risk perception.

As 2023 heralds the third year of COVID-19 vaccination rollout, it is crucial to assess progress made by countries in the African region in 2022, given substantial efforts put in by the WHO AFRO and other partners in low-performing countries. This study summarizes the progress made in 2022 in rolling out COVID-19 vaccination and analyses factors associated with vaccination coverage in 2022 in the general population in the African region.

## 2. Materials and Methods

A retrospective analysis of COVID-19 vaccine uptake as of the end of 2022 in the African region and associated factors was conducted.

### 2.1. Inclusion and Exclusion Criteria

We included countries of the African region that reported COVID-19 vaccine uptake data to the WHO AFRO from January 2021 to December 2022.

Countries that had not started to roll out COVID-19 vaccination as of 31 December 2022, as well those that did not formally submit reports on COVID-19 vaccine uptake, including reports of zero doses administered or zero individuals vaccinated during the last 3 months, were excluded. Only Eritrea met the exclusion criteria.

### 2.2. Data Sources and Measurement

The regional database on COVID-19 vaccinations maintained by the WHO AFRO from reports submitted by Member States between 13 January 2021 and 31 December 2022, was used for analysis. In addition, a second dataset was built using data reported on vaccination’s operations and socio-economic factors from the literature. This included the following variables: country, population, universal health service coverage, income classification, whether the country benefits from the WHO AFRO support as part of the MP–CST initiative, number of days of support from international experts deployed by WHO AFRO in 2022, implementation of high-volume mass vaccination campaigns, WHO funding spent in 2022 per person who had completed the primary vaccination series, and number of days spent implementing COVID-19 vaccination programs in 2022. We also used the line list of COVID-19 cases and deaths maintained by the WHO AFRO based on Member States’ notification as of 31 December 2022, as per their obligations under the International Health Regulations (2005) [15] and the Integrated Disease Surveillance and Response Strategy (IDSR) [16].

### 2.3. Data Analysis

Using the dataset derived from all collected data, the following analyses were performed:(a)Vaccine uptake and coverage cumulatively and during the years 2021 and 2022.

For each country, we computed the following parameters:
Percentage of doses administered over received: The cumulative doses administered was divided by the cumulative doses received and multiplied by 100.Percentage of doses expired over received: The cumulative doses expired was divided by the cumulative doses received and multiplied by 100.Percentage of people who had completed the primary vaccination series ((initial number of doses of a particular vaccine that a person needs), i.e., one dose of the Janssen vaccine or two doses of other vaccines [11]) in the general population and in high-priority risk groups (using healthcare workers as a tracer): The cumulative number of people having completed the primary series was divided by the general population or the population size of the related high-risk group, and multiplied by 100. Booster doses were not included in the primary series definition [11].Percentage of people who have completed the primary series and received at least one booster dose: The cumulative number of people who received the first dose of booster was divided by the number of people who have completed the primary series and multiplied by 100.


(b)Multivariable analysis of factors associated with COVID-19 vaccination coverage in 2022.


The outcome of interest was the percentage of people who had completed the primary vaccination series in 2022. The following variables were considered as covariates: population, MP–CST country, income classification, universal health service coverage index, percentage of people who completed the primary series and received the Janssen vaccine, number of cases per million population in 2022, number of deaths per million population in 2022, implementation of at least one high volume mass vaccination campaign in 2022 (resulting in at least 5% of people having completed the primary vaccination series), number of days of support from international experts deployed by WHO AFRO in 2022, and WHO funding spent in 2022 per person who completed the primary series. At the end of 2021, the five countries that surpassed 40% of people with complete primary series had a population size below the region median, while fewer than 5% of people in the four most populated countries (United Republic of Tanzania, Democratic Republic of Congo, Ethiopia, Nigeria) had completed the primary series. Accordingly, country population size was selected among covariates. Income classification and the universal health service coverage index were used as variables related to a health system’s capacity to deploy COVID-19 vaccines. Given the limited capacity of most countries, the fact of being a priority country that benefited from intensive external assistance through the MP–CST initiative, was also included among covariates. Assuming that an increase in the number of new cases and deaths may boost vaccine demand, the number of cases per million population and the number of deaths per million population in 2022 were included among covariates. The fact of having implemented at least one high-volume mass vaccination campaign in 2022 or not was selected as a covariate for vaccine delivery strategies. The World Bank and Gavi are the main sources of external funding for the deployment of COVID-19 vaccines in most countries in the African region. WHO funding spent per person who completed the primary series in 2022 was included among covariates as a tracer for the utilization of external funds to support the deployment of COVID-19 vaccines.

The countries that achieved 70% of people having completed the primary series by the end of 2021 (Seychelles and Mauritius) were excluded in the multivariable analysis, as they were not expected to intensify COVID-19 vaccines rollout efforts in 2022.

Due to the method’s flexibility in allowing for overdispersion, risk factors for COVID-19 vaccination coverage were fitted using the negative binomial regression. The likelihood ratio test was used to test this assumption.

Both univariable and multivariable regression models were fitted. As a result of the univariable analysis, all covariates with a *p*-value < 0.25 [17] were included in the initial multivariable model. A backward stepwise method was used with gradual deletion of variables with *p* ≥ 0.25. The strength of the association between each covariate and the dependent variable was measured by the β coefficients with a 95% confidence interval. The likelihood ratio test was used to test the goodness of fit of the final model.

Data were analyzed using R version 4.2.1 [18] for statistical analysis and ESRI 2017 ArcGIS Pro 2.1.0 [19] for mapping.

## 3. Results

### 3.1. COVID-19 Vaccines Doses Received and Administered

In 2022, 448,942,809 COVID-19 vaccine doses were delivered in the African region; this figure represents a 39.9% increase compared to the 320,841,531 vaccines delivered in 2021. Cumulatively, as of 31 December 2022, 769,784,340 doses were delivered; 66.5% originated from the COVID-19 Vaccines Global Access (COVAX) facility (n = 512,089,328), 20.7% from bilateral donations (n = 159,555,491), and 9.4% from the African Vaccine Acquisition Trust (AVAT) (n = 72,236,700); a further 3.4% were purchased directly by governments or originated from unspecified sources (n = 25,902,821). Janssen, Pfizer-BioNTech and Sinopharm vaccines accounted for 37.0%, 18.8% and 13.3% of doses received, respectively. Doses received as of 31 December 2022 represented 47.0% of the doses needed to provide the complete primary vaccination series to 70% of people in all countries. Out of the 47 countries in the African region, only Eritrea has not received COVID-19 vaccines.

Of doses received, 70% (n = 541,185,838) had been administered by 31 December 2022 against 53% by 31 December 2021. A total of 370,312,710 doses were administered in 2022, compared to 170,873,128 doses in 2021 (a 117% increase). As of 31 December 2022, 13 countries had administered fewer than 50% of vaccine doses received, compared to 24 countries at the end of December 2021.

Of the 46 vaccine-receiving countries, 35 have reported expired COVID-19 vaccines to the WHO AFRO. As of 31 December 2022, 23,511,577 million expired doses had been recorded, representing 3.1% of doses received in the African region. The median percentage of expired doses over those received by country was 2.8%, ranging from 0.1% to 28.6%. Algeria, Senegal, Madagascar, the Democratic Republic of Congo, and Nigeria accounted for 65% of the expired doses reported in the African region.

### 3.2. COVID-19 Vaccination Coverage 

As of 31 December 2022, 308.1 million people have completed the primary vaccination series, representing 26.4% of the region’s population; a further 373.4 million people have received at least one dose of a COVID-19 vaccine (32.0% of the region’s population). The percentage of people who had completed the primary series increased from 6.3% at the end 2021 to 26.4% at the end 2022 (Figure 1).

At the end of December 2021, only Mauritius and Seychelles had vaccinated at least 70% of their population; they were joined by Liberia and Rwanda by the end of December 2022. Burundi, Democratic Republic of Congo, Madagascar, and Senegal had yet to achieve 10% coverage by the end of 2022 (Figure 2).

The following countries recorded the highest percentages of the people who had completed the primary series during 2022: Liberia (64.0%), United Republic of Tanzania (46.0%), Sierra Leone (43.2%), Zambia (41.7%), Mozambique (41.3%), Cote d’Ivoire (36.9%), and Ethiopia (30.5%). Vaccination coverage stagnated in 2022 in Algeria, Burundi, Equatorial Guinea, Congo, Senegal, and Gabon with 3% or fewer people completing the primary series during the year 2022 (Figure 3 and Figure 4).

In total, 37 countries reported on booster shots to the WHO AFRO, while 9 countries had either not included booster doses in their COVID-19 vaccination schedule or were not reporting on booster doses. The proportion of people who had completed primary series and received at least one booster dose was 17.3% (n = 44,106,608). Figure 5 shows the distribution of this parameter by country in the African region as of 31 December 2022.

Health workers were used as a tracer for all high-priority risk groups. Overall, 24 countries reported data on the number of health workers who had completed the primary series. The average percentage of health workers with complete primary series was 40.9%, ranging from 11.6% in Burundi to 99.4% in Rwanda (Figure 6).

### 3.3. Factors Associated with Vaccine Coverage in 2022

Forty-four countries were considered for the multivariable analysis (excluding Eritrea, Seychelles, and Mauritius).

Table 1 shows the association between the percentage of people with who had completed the primary series in 2022 and potential risk factors using the non-adjusted and adjusted negative binomial regression analyses. Having carried out at least one high volume mass vaccination campaign in 2022 was associated with high vaccination coverage (β = 0.91, *p* < 0.0001), while higher WHO funding spent per person vaccinated (complete primary series) in 2022 was associated with low vaccination coverage (β = −0.26, *p* < 0.03).

In total, 17 countries out of 46 in the African region implemented at least one high-volume mass vaccination campaign: Burkina Faso, Cameroon, Chad, Cote d’Ivoire, Ethiopia, Guinea, Kenya, Liberia, Mozambique, Niger, Nigeria, Rwanda, Sao Tome and Principe, Sierra Leone, South Sudan, Uganda, United Republic of Tanzania, and Zambia.

No significant association was found between vaccination coverage in 2022 and the universal health coverage services index, the country population, the income classification, the number of cases reported in 2022 per million population, the number of deaths reported in 2022 per million population, and the fact of being a priority country for the WHO AFRO (benefiting from the MP–CST initiative).

## 4. Discussion

While 63% of the global population have completed the primary vaccination series for COVID-19 at end 2022 [20], vaccination coverage remains low in the African region. Less than 26% of the population has completed their primary series and only four countries have reached the 70% coverage target set by the WHO for all countries by mid-2022 [11]. The first eight months of 2021 represent a missed opportunity for COVID-19 vaccine uptake in Africa due to limited vaccines availability during two major waves of the pandemic, driven by the Delta and Omicron variants of concern [21]. The Delta wave would have been a prime opportunity to capitalize on a surge in vaccine demand as a result of increased risk perception, which was caused by a surge in hospitalizations and deaths in many countries in the African region [22]. At that time, developed countries were starting to administer booster doses while Africa still struggled to vaccinate its healthcare workers [23]. However, from August 2021 onward, vaccine supply in the African region improved significantly and ceased to be a reason for low vaccine uptake [10]. Tagoe et al. [24], working in Ghana and Bangladesh, found that challenges to the COVID-19 vaccination rollout in low- and middle-income countries included vaccine hesitancy, low resource availability, poor roads on which to transport vaccines, inadequate cold chain storage, lack of coordination with the private healthcare sector, gender differences, and limited political commitment.

In 2022, several countries in the African region scaled up COVID-19 vaccination, leading to a significant increase in vaccination coverage (complete primary series). In this study, conducting at least one high-volume mass vaccination campaign in 2022 was associated with high vaccination coverage. Mass vaccination involves delivering vaccines to a large number of people at one or more locations in a short interval of time [25]. This usually involves deploying outreach and mobile teams in addition to fixed health facility vaccination sites. In the African region, this vaccine delivery strategy has been used by some countries, such as Liberia, Ethiopia, Cote d’Ivoire, United Republic of Tanzania, and Zambia, to rapidly increase COVID-19 vaccination coverage. During mass vaccination campaigns, activities aiming at generating vaccines demand are usually intensified to counter vaccine hesitancy, which is considered the leading cause of low vaccine uptake in Africa [21]. Lawal et al. [23] found that the high hesitancy in most African countries is influenced by mistrust of government programs resulting from a legacy of poor service delivery, as well as lack of confidence in vaccines’ efficacy, the integrity of the providers; complacency and anti-vaccine campaigns on social media. Several other authors have documented vaccine hesitancy as one of the factors associated with low vaccine uptake in Africa [26,27,28]. Kabakama et al. [29] concluded that political influences, religious beliefs, and low risk perception contribute to COVID-19 vaccine hesitancy in 12 sub-Saharan countries. It is critical for governments, policymakers, community leaders and health workers at all levels in the African region to understand the scientific basis for COVID-19 vaccination, set up or strengthen plans to counter rumors, adequately explain the facts, build trust, and invest more in engaging communities in vaccine demand generation activities [29]. National responses to the COVID-19 pandemic should also generate their own evidence on vaccine safety and efficacy. For this purpose, local data on the vaccination status of new COVID-19-related admissions to health facilities, and transparent documentation of any recorded severe adverse events following immunization, are key to building public trust in COVID-19 vaccines.

In this study, higher WHO funding spent per person who had completed the primary vaccination series in 2022 was associated with low vaccination coverage. WHO funding, allocated to Member States based on agreed yearly work plans and through WHO Country Offices, is meant to be catalytic funds and to cover existing gaps. The core funding for COVID-19 vaccination rollout was provided by Governments, GAVI (through the COVID-19 Vaccine Delivery Support (CDS) funding), the World Bank [30], and other partners. This negative association may mean that WHO funding was mostly used by countries with the highest gap in funding for COVID-19 vaccine rollout. Idris IO et al. [31] identified operational gaps among the challenges of vaccines rollout in Africa, while Lawal et al. [23] identified low resource availability and inadequate cold chain storage among factors driving low vaccine uptake, pointing to already fragile health systems, which were further stressed by the COVID-19 pandemic. Several countries in the African region lack the competence, logistic capabilities (including ultra-cold chain management), financial resources, and sufficient health workforce to mass-vaccinate their populations [23]. In addition to the delivery of vaccines with short shelf lives, limited capacity to rollout COVID-19 vaccination has led to COVID-19 vaccines expiring. In this study, 3.1% of doses received in the African region expired. This highlights the need to adjust the distribution of vaccines to fixed, outreach and mobile vaccination sites based on demand to prevent open vial wastage. There’s also a need to improve the tracking of vaccine expiry dates by batch to prevent closed vial wastage [32], and to effectively apply recommendations from the WHO, Africa Center for Disease Control, UNICEF and GAVI joint statement on the donation of COVID-19 vaccines to African countries [33]. This joint statement highlights the need for a predictable and reliable supply, and a shelf life of a minimum of ten weeks when vaccines arrive in the designated country. Indeed, having to plan at short notice and ensure uptake of doses with short shelf lives exponentially magnifies the logistical burden on health systems that are already stretched in most African countries [33].

In this study, no association was found between COVID-19 vaccination coverage and either the number of cases per million population or the number of deaths per million population in 2022. Indeed, in 2022, there was a considerable reduction in risk perception in most countries in the African region, as a result of low incidence and death rates; declining risk perception prevented increased vaccine demand. This may be attributed to the displacement of the Delta variant with the less virulent but highly transmissible Omicron variant [34]. Despite the reduction in deaths, there is still a need to protect vulnerable groups through the achievement of high vaccination coverages and herd immunity [34]. In a cross-sectional study conducted in Algeria between 23 December 2021 and 12 March 2022, Hamimes et al. [35] found that the role of preventive measures (social distancing, quarantine, and facemask wearing) was limited and insufficient for mitigating the spread of COVID-19 when people are unvaccinated; this finding highlights the importance of combining preventive measures and vaccination to control the COVID-19 pandemic.

Limited political engagement and will to promote vaccination has also hindered COVID-19 vaccination in some African countries [10]. Eritrea has not yet introduced COVID-19 vaccines as part of its national response to the pandemic. Similarly, Burundi and Tanzania delayed deploying COVID-19 vaccines until their political leadership made a major shift to embrace global public health measures and include a vaccination pillar in the national COVID-19 response plans, in July and September 2021, respectively [36,37]. Achieving and maintaining high levels of vaccination coverage require commitment to immunization by various actors from communities to the national level. The primary responsibility, however, falls on governments, which must commit to developing and delivering vaccines within their primary healthcare systems and ensuring the quality of vaccination services [38].

Despite the overall low coverage, COVID-19 vaccination has saved lives in the African region. Watson et al. [5] estimated that 466,400 [446,300;487,000] deaths were averted by COVID-19 vaccination in the African region as of the end of 2021. Consequently, the number of lives saved by COVID-19 vaccination markedly exceeded the reported death toll as of the end of 2021 (n = 154,774) [39,40].

The number of new cases and deaths reported in 2022 remained very low in most countries in the African region, despite the overall low vaccination coverage. This is in stark contrast to the situation seen in China in the last quarter of 2022, where poorly vaccinated populations had been exposed to the Omicron variant, without previous exposure to earlier variants, leading to an upsurge in COVID-19 cases and deaths [41]. This suggests that high exposure to the virus during subsequent waves of the pandemic over the past three years led to increased population immunity. There are several studies on SARS-CoV-2 seroprevalence [42] that support the view that a very high proportion of the African region had SARS-CoV-2 exposure. However, increasing COVID-19 vaccination coverage is still necessary across the continent as the virus is still circulating, and the emergence of more virulent variants cannot be ruled out. Learning from the first 18 months of the COVID-19 vaccination rollout, the WHO, in July 2022, updated its global COVID-19 vaccination strategy to reflect the changing world [43]. In this new strategy, WHO recommended that countries progress towards an aspirational target of vaccinating 100% of healthcare workers, older adults and other high-risk groups with the primary vaccination series and booster doses. This study has shown that progress made in vaccinating high risk groups is not fully known for all countries in the African region. Only 24 countries reported data on health workers who completed the primary vaccination series. In these 24 countries, 40.7% of healthcare workers completed the primary vaccination series. The disproportionate coverage for healthcare workers between countries, evidenced by the large range from 11.2% to 99.4%, may indicate a population size issue that may have led to over- or under-estimation of vaccination coverage in high-risk groups. It is critical for countries in the African region, in line with the updated WHO global COVID-19 vaccination strategy [43], to adjust not only the population size estimates for high-risk groups, but also their delivery strategies. The focus should be first on vaccinating high-risk groups, using the Provider Initiated approach in health facilities and on integrating COVID-19 vaccination services into routine care for those living with HIV, tuberculosis, cancer, diabetes and other noncommunicable diseases, as well as into home care services and housing for elderly people [10].

### Limitations

In this study, the vaccination coverage was estimated using data reported by Member States. Some countries are experiencing backlogs in data reporting at country level and/or delays in data submission to the WHO AFRO. The data used in this study may not be complete, leading to under-estimation of vaccination coverage in some countries. In addition, population sizes for health workers provided by Member States may not be accurate in some countries, leading to over- or under-estimation of vaccination coverage for the high-risk groups.

Regarding some covariates used in the multivariable analysis, for which data were collected from available sources, the most recent available data were used; some of these data were not from the same year as the values used to calculate vaccination coverage. Some data used were sourced from publicly available repositories and therefore subject to the limitations of their sampling, study design and collection processes.

The interpretation of the results of this study should take these limitations into account.

## 5. Conclusions

The COVID-19 pandemic is still ongoing in the African region despite very low-reported incidences in most countries in the past few months. In 2022, most countries in the African region scaled up COVID-19 vaccination programs, resulting in significant improvement in vaccination coverage. Despite commendable efforts and progress, overall vaccination coverage remains low in the African region, including among high-risk groups. This could delay full recovery from the COVID-19 pandemic. The results of this study showed the positive role of high-volume mass vaccination campaigns in increasing vaccination coverage. However, mass vaccination campaigns are cost-intensive, implemented as an emergency response and are often funded through African countries’ health budgets. Given its relatively high cost, African countries cannot sustainably use only this strategy to achieve their vaccination goals, such as the vaccination of most high-risk groups. It is essential that countries integrate COVID-19 vaccination into routine immunization and primary healthcare, and continue to invest more in engaging community leaders in demand creation activities during the transition period that follows the acute phase of the pandemic. Countries’ capacity to continue generating demand for COVID-19 vaccines in a context of reduced risk perception is key to controlling the pandemic. In addition, ensuring that most high-priority groups complete their primary vaccination series and receive periodic boosting as immunity wanes is vital to controlling the pandemic, preventing the emergence of new variants of concern, reducing any residual threats to health systems, and to continuing economic activities.

## Figures and Tables

**Figure 1 vaccines-11-01010-f001:**
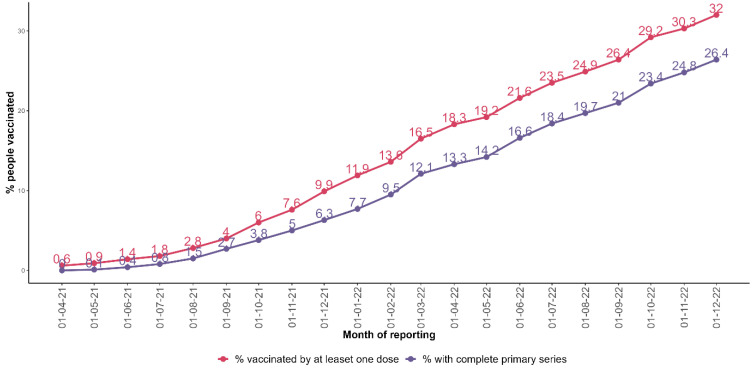
Percentage of people who had received at least one COVID-19 vaccine dose and people who had completed the primary series by reporting month in the African region.

**Figure 2 vaccines-11-01010-f002:**
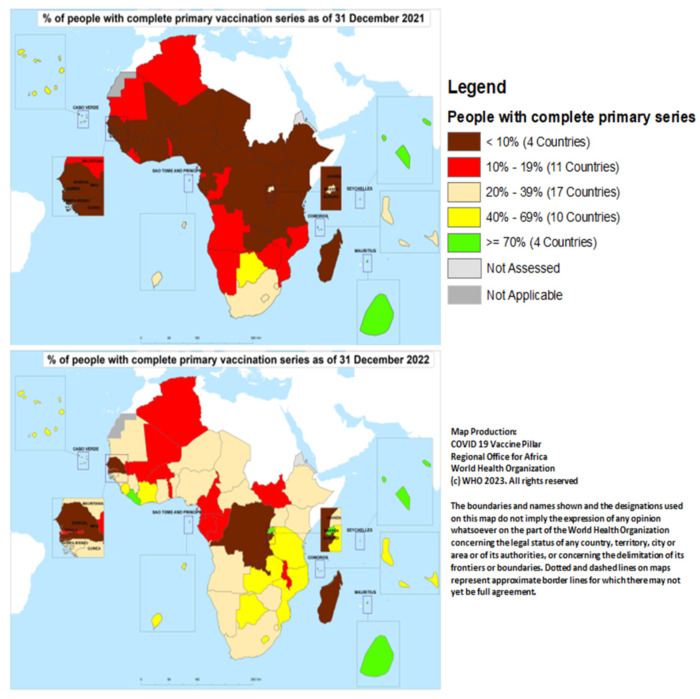
Proportion of people who had completed the primary COVID-19 vaccination series by country in the African region at the end of December 2021 and 2022.

**Figure 3 vaccines-11-01010-f003:**
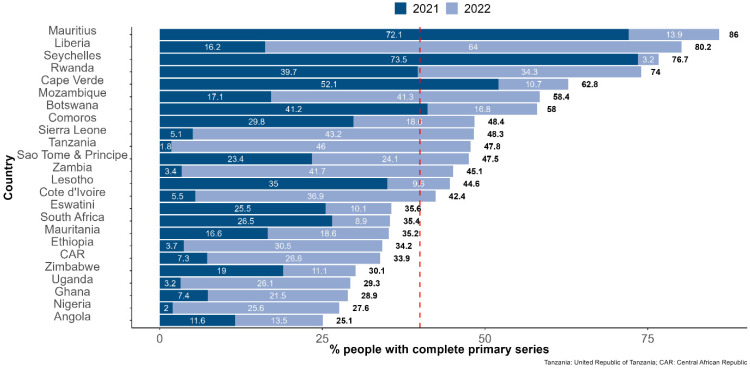
Percentage of people who had completed the COVID-19 vaccination primary series at the end of December 2021 and 2022 among countries with more than 25% of people who had completed the primary series as of 31 December 2022. Note: The dashed red line highlights the 40% of people having completed the primary series mark, set as a target for the end of 2021 by the WHO.

**Figure 4 vaccines-11-01010-f004:**
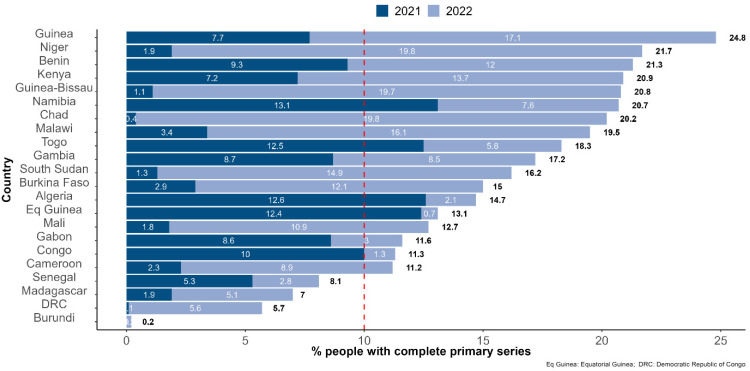
Percentage of people who had completed the COVID-19 vaccination primary series by the end of December 2021 and 2022, respectively, among countries with fewer than 25% of people having completed the primary series as of 31 December 2022. Note: The dashed red line highlights the 10% of people having completed the primary series mark.

**Figure 5 vaccines-11-01010-f005:**
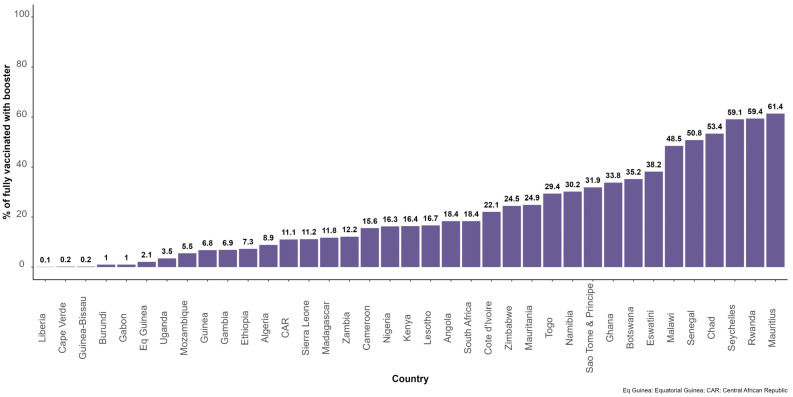
Proportion of people with who had completed the COVID-19 vaccination primary series and received booster doses in 35 countries in the African region.

**Figure 6 vaccines-11-01010-f006:**
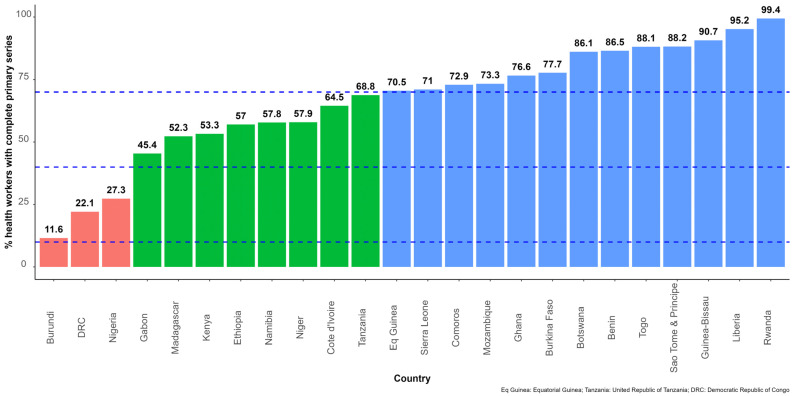
Percentage of health workers who had completed the primary COVID-19 vaccination series in 24 countries in the African region. Note: The dashed blue lines highlight the 10%, 40% and 70% of people having completed the primary series marks.

**Table 1 vaccines-11-01010-t001:** Multivariate analysis of factors associated with vaccination coverage in 2022 in the 46 countries of the African region.

Explanatory Variables	Mean ± sd or n (%)	Crude Beta Coefficient			Adjusted Beta Coefficient (Final Model)		
		Value	95% CI	*p*	Value	95% CI	*p*
Population (million)	264.5 ± 390.7	0.0002	[−0.0003; 0.0009]	0.50			
Income classification (Middle income)	24 (54.5%)	−0.32	[−0.80; −0.15]	0.18	−0.0026	[−0.039; 0.34]	0.89
UHC service coverage index	46.5 ± 10.0	−0.015	[−0.04; 0.012]	0.23			
Number of cases per million population in 2022	3745 ± 9125	−9.14 × 10^−6^	[−3.25 × 10^−5^; 2.04 × 10^−5^]	0.50			
Number of deaths per million population in 2022	231.0 ± 414.1	−0.0005	[−0.001; 0.0001]	0.11	−0.00042	[−0.00054; 0.00048]	0.86
% of people with complete primary series who received Janssen vaccine	51.63 ± 33.16	0.004	[−0.004; 0.011]	0.32			
MP–CST country (Yes)	19 (43.18%)	0.47	[0.004; 0.94]	0.049	−0.0078	[−0.47; 0.314]	0.69
Days of expertise provided by WHO AFRO	263 ± 376	0.0002	[−0.0005; 0.0009]	0.57			
Implemented at least one high volume mass vaccination campaign (Yes)	18 (40.91%)	1.03	[0.66; 1.41]	<0.001	0.91	[0.52; 1.31]	<0.0001
WHO funding spent per people vaccinated (complete primary series) in USD	0.88 ± 2.08	−0.35	[−0.59; −0.16]	0.001	−0.26	[−0.46; −0.11]	0.03

MP–CST: Country Support Team. CI: Confidence Interval. Likelihood ratio test: χ2 = 2.65, *p* = 0.10.

## Data Availability

The data presented in this study are available from the corresponding author upon request.

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
