# Peer review of "COVID-19 Vaccination in the WHO African Region: Progress Made in 2022 and Factors Associated"

_vaccines, 2023, doi:10.3390/vaccines11051010_

Round 1
Reviewer 1 Report
In this paper, the authors evaluate drivers of vaccine uptake within African countries in 2022 using a regional database on COVID-19 vaccination maintained by the WHO AFRO and a second dataset built on data reported on vaccine response and socio-economic factors.
1. Some English grammar updates needed (missing articles etc)
2. Would define what was considered the primary vaccine series
3. Where were the Seychelles and Mauritius not in the MV analysis?
4. Is the word "lower" missing? Though later on seems like it means any funding?: LOWER WHO funding spent per people vaccinated (complete primary series) in 2022 was associated with low vaccination coverage (β = -0.26, P < 0.03).
Would add more to the current explanation in the discussion.
5. Figure 2: Not clear where the not applicable color is used.
6. Figure 3: Would add what the red line is
7. Figure 5: Would have the y axis to 100%.
8. Figure 6: Would explain the blue lines
9. Not sure what this sentence means. "During mass vaccination campaigns demand generation activities are intensified to counter vaccine hesitancy which is considered the leading cause of low vaccine uptake in Africa"
10. Page 11, Line 323: Is lack of reporting also a potential reason for low case numbers?
11. Was a univariate analysis conducted before the multivariate?
Author Response
- Some English grammar updates needed (missing articles etc)
RESPONSE: Thank you for this relevant comment. We have reviewed and edited the manuscript.
- Would define what was considered the primary vaccine series
RESPONSE: Thank you for this relevant comment. The definition has been added the first time “primary vaccination series” was used in the Method section (see lines 111-112).
- Where were the Seychelles and Mauritius not in the MV analysis?
RESPONSE: Thank you for this comment. Mauritius and Seychelles were excluded from the Multivariable analysis because both countries surpassed 70% of people with complete primary series at the end of 2021, and were not expected to intensify efforts in rolling out the COVID-19 vaccine in 2022 (see lines 150-152).
- Is the word "lower" missing? Though later on seems like it means any funding?: LOWER WHO funding spent per people vaccinated (complete primary series) in 2022 was associated with low vaccination coverage (β = -0.26, P < 0.03).Would add more to the current explanation in the discussion.
RESPONSE: Thank you for this comment. This is a negative association. So the appropriate word is “higher” and not “lower” WHO funding spent per person vaccinated (associated with low vaccination coverage). This has been corrected in the summary, results, and discussion sections.
- Figure 2: Not clear where the not applicable color is used.
RESPONSE: Thank you for this comment. Not applicable was used for the Western Sahara territory. This is in line with WHO’s Standard Operational Procedures on GIS.
- Figure 3: Would add what the red line is.
RESPONSE: Thank you for this suggestion which has been addressed.
- Figure 5: Would have the y axis to 100%.
RESPONSE: Thank you for this suggestion which has been addressed.
- Figure 6: Would explain the blue lines
RESPONSE: Thank you for this suggestion which has been addressed.
- Not sure what this sentence means. "During mass vaccination campaigns demand generation activities are intensified to counter vaccine hesitancy which is considered the leading cause of low vaccine uptake in Africa".
RESPONSE: Thank you for this comment. This sentence has been rephrased as follows: "During mass vaccination campaigns, activities aiming at generating vaccine demand are usually intensified to counter vaccine hesitancy which is considered the leading cause of low vaccine uptake in Africa" (see lines 295-297).
- Page 11, Line 323: Is lack of reporting also a potential reason for low case numbers?
RESPONSE: Thanks for this suggestion. We have addressed this in the text (see line 337).
- Was a univariate analysis conducted before the multivariate?
RESPONSE: Thanks for this comment. We can confirm that a univariate analysis was conducted before the multivariate analysis. This was indicated in lines 157-159. In Table 1, the Crude Beta coefficient was computed from the univariate analysis.
Reviewer 2 Report
The objectives of the study have been defined well defined.
The methodology for data collection and analysis has been described adequately.
The authors have analyzed data on Covid -19 vaccination during the early and late pandemic.Factors influencing the vaccination process have been discussed with appropriate statistical analysis and graphic representation of the trends in various African countries. The reasons for the trends have been analyzed adequately
Two comments to be addressed:
1. The reasons for acquiring vaccines post-expiry date may be highlighted. This will reflect on the procedural or logistic failure of delayed clearance for receiving the vaccines, reflecting strategies for correction.
2. WHO funding procedure to facilitate vaccines may be elaborated so as to understand the impact of low vaccination.

Author Response
1) The reasons for acquiring vaccines post-expiry date may be highlighted. This will reflect on the procedural or logistic failure of delayed clearance for receiving the vaccines, reflecting strategies for correction.
RESPONSE: Thank you for this relevant suggestion. We believe the Reviewer was referring to vaccines with short shelf-life instead of vaccines with post-expiry dates. In the descriptive part of this manuscript, we presented an update on cumulative doses that expired in the African region (lines 183-188). The reasons for the expiry of doses as well as the logistical implications were discussed from lines 320 to 330. We have added a paragraph highlighting the need to effectively apply recommendations from the WHO, Africa Center for Disease Control, UNICEF, and GAVI joint statement on the donation of COVID-19 vaccines to African countries [Ref] related to the need for a predictable and reliable supply, and a minimum of ten weeks shelf life as vaccines arrive in a country (see lines 324-330).
2) WHO funding procedure to facilitate vaccines may be elaborated so as to understand the impact of low vaccination.
RESPONSE: Thank you for this relevant comment. We have specified the following on lines 306-309:
“In this study, higher WHO funding spent per person having completed the primary vaccination series in 2022 was associated with low vaccination coverage. WHO funding, allocated to Member States based on agreed yearly work plans, through WHO Country Offices, is meant to be catalytic and to cover existing gaps”
Reviewer 3 Report
The paper describes and discusses the COVID-19 vaccination progress made by the African region in 2002. The paper includes both a descriptive analysis of the vaccination rollout across different African countries in 2022, and a regression analysis of the factors associated with the rollout. I found the descriptive analysis very thorough and enlightening. On the other hand, I don’t understand i) what the purpose of the regression analysis is, and ii) what we are actually learning from it. The paper does not offer any clear guidance of why those specific covariates were included in the regression (apart from being significatively correlated with vaccination rates in a univariate model). For instance, why would you expect the universal health service coverage to be associated with COVID-19 vaccination coverage? What about the days of expertise provided by WHO AFRO? The paper also does not provide any policy implications stemming from the regression analysis.
The paper does not provide a comprehensive discussion of the potential factors associated with COVID-19 vaccination rates. Instead, it focuses on a limited set of measurable factors. Generally speaking, I think there are many reasons explaining differences in vaccination rates across countries, both on the demand side and on the supply side. Some of the factors on the demand side are trust in the government or the healthcare system, confidence in the vaccine efficacy, and perceived risk of COVID-19 infection or of the vaccine side effects. Some of the factors on the supply side are infrastructure and logistics (e.g., number of healthcare workers, number of people living in hard-to-reach areas), number of doses available, the cost of distributing the doses compared to the overall health budget.
Figure 2 is difficult to understand. In particular, it is not clear what the three graphs represent.
Author Response
1) I don’t understand i) what the purpose of the regression analysis is, and ii) what we are actually learning from it.
RESPONSE: Thank you for this comment. The purpose of the multivariable analysis is to analyze factors associated with vaccination coverage in 2022 in the general population. Indeed, at the end of 2021, most countries missed the target set by WHO, with 26 countries out of 46 having less than 10% of their population with complete primary vaccination series. In 2022, several countries scaled up the COVID-19 vaccination rollout resulting in a significant increase in vaccine coverage. We wanted to identify drivers of vaccination coverage. From this analysis, we learned the following:
- a) conducting at least one high-volume mass vaccination campaign in 2022 was associated with high vaccination coverage.
- b) WHO funding spent per person having completed the primary vaccination series in 2022 was associated with low vaccination coverage. This negative association may mean that WHO funding was mostly used by countries with the highest gap in funding for COVID-19 vaccine rollout as core funding is provided by Governments, Gavi, and the World Bank.
c) Vaccination coverage in 2022 was not driven by the universal health coverage services index or income classification. Given that vaccines were provided for free to countries through COVAX Facility, AVAT, and bilateral donations, and the availability of external funding to support vaccination rollout, countries’ income classification and the capacity of the health system did not have a significant impact on COVID-19 vaccination coverage in 2022. - d) Despite the fact that most of the countries that have surpassed 70% of people with complete primary series have small population sizes, no association was found between vaccination coverage in 2022 and the country’s population size.
2) The paper does not offer any clear guidance of why those specific covariates were included in the regression (apart from being significantly correlated with vaccination rates in a univariate model). For instance, why would you expect universal health service coverage to be associated with COVID-19 vaccination coverage? What about the days of expertise provided by WHO AFRO?
RESPONSE: Thank you for this relevant comment. The rationale for the selection of covariates has been added in the methods section (see lines 132-149).
3) The paper also does not provide any policy implications stemming from the regression analysis.
RESPONSE: Thank you for this comment. The policy implications of the multivariable analysis are highlighted in the conclusion section (see lines 416-425).
4) The paper does not provide a comprehensive discussion of the potential factors associated with COVID-19 vaccination rates. Instead, it focuses on a limited set of measurable factors. Generally speaking, I think there are many reasons explaining differences in vaccination rates across countries, both on the demand side and on the supply side. Some of the factors on the demand side are trusted in the government or the healthcare system, confidence in the vaccine efficacy, and perceived risk of COVID-19 infection or of the vaccine side effects. Some of the factors on the supply side are infrastructure and logistics (e.g., number of healthcare workers, number of people living in hard-to-reach areas), number of doses available, and the cost of distributing the doses compared to the overall health budget.
RESPONSE: Thank you for this comment. In the discussion section, the association between the vaccination rate in 2022 and high-volume mass vaccination campaigns was discussed between lines 287 and 294. Given that we could not get data on vaccine demand for most countries, factors related to vaccine demand were not included as covariates. However, considering that, during mass vaccination campaigns demand generation activities are intensified to counter vaccine hesitancy, the link between vaccination coverage and vaccine demand was discussed (see lines 287-298).
The association between WHO funding spent per person vaccinated and vaccination coverage in 2022 was discussed between lines 306 and 313. This provided the opportunity to raise the issue of low resource availability and inadequate cold chain storage (see lines 313-318).
The fact that there was no association between COVID-19 vaccination coverage and the number of cases per million population as well as the number of deaths per million population in 2022 was discussed between 331 and 347.
5) Figure 2 is difficult to understand. In particular, it is not clear what the three graphs represent.
RESPONSE: Thank you for this relevant comment. We have deleted the 2022 map to keep the ones presenting the distribution of the cumulative vaccination coverage by in country as of end 2021 and end 2022, for better understanding.
Round 2
Reviewer 3 Report
The authors have addressed most of my comments or provided an explanation for not addressing them.
Author Response
Reviewer comment: English language and style are fine/minor spell check required.
RESPONSE: Thank you for this relevant comment. We have reviewed and edited the manuscript.